# Impact of Transcriptome and Gut Microbiome on the Response of HIV-1 Infected Individuals to a Dendritic Cell-Based HIV Therapeutic Vaccine

**DOI:** 10.3390/vaccines9070694

**Published:** 2021-06-24

**Authors:** Roque Pastor-Ibáñez, Francisco Díez-Fuertes, Sonsoles Sánchez-Palomino, Jose Alcamí, Montserrat Plana, David Torrents, Lorna Leal, Felipe García

**Affiliations:** 1AIDS Research Group, IDIBAPS, Hospital Clinic, University of Barcelona, 170, 08036 Barcelona, Spain; rpastor@clinic.cat (R.P.-I.); ssanchez@clinic.cat (S.S.-P.); alcami@clinic.cat (J.A.); mplana@clinic.cat (M.P.); fgarcia@clinic.cat (F.G.); 2Instituto de Salud Carlos III, Ctra. de Pozuelo, 28, Majadahonda, 28222 Madrid, Spain; frandifu@icloud.com; 3Computational Genomics Groups, Barcelona Supercomputing Center (BSC), Plaça d’Eusebi Güell, 1-3, 08034 Barcelona, Spain; david.torrents@bsc.es; 4Infectious Diseases—Department, Hospital Clínic, IDIBAPS, University of Barcelona, Villarroel, 170, 08036 Barcelona, Spain

**Keywords:** microbiota, transcriptomics, dendritic cell, vaccine, antiretroviral, HIV-1

## Abstract

Therapeutic vaccines based on dendritic cells offer a good approach to HIV-specific T-cell responses and partial control of the viral load after antiretroviral therapy interruption. The aim of the present study was to identify mRNA expression profiles and to assess the impact of the gut microbiome composition for predicting the viral load control after antiretroviral therapy interruption. We enrolled 29 patients to receive either placebo or a monocyte-derived dendritic cell vaccine. Patients with a decrease in their viral load of >0.5 log_10_ copies/mL by 12 weeks after antiretroviral therapy interruption were considered responders. In total, 66 genes were considered differentially expressed between responders and non-responders. Enrichment analysis revealed several upregulated pathways involved in the host defense response to a virus via the type I interferon signaling pathway. Regarding the gut microbiota, responders showed enriched levels of Bacteroidetes (*p* < 0.005) and Verrucomicrobia (*p* = 0.017), while non-responders were enriched with Tenericutes (*p* = 0.049) and Actinobacteria (*p* < 0.005). We also found important differences at the genus level. However, we did not discover any effect of the dendritic cell vaccine on the transcriptome or the gut microbiota. An alternative analysis did characterize that the microbiota from responders were associated with the metabolic production of short-chain fatty acids, which are key metabolites in the regulation of intestinal homeostasis. The evidence now consistently shows that short-chain fatty acid depletion occurs in HIV-infected individuals receiving antiretroviral treatment.

## 1. Introduction

Therapeutic vaccines are a promising strategy for HIV remission and cure. The main goal is to boost the immune system of infected individuals to achieve control of the viral replication without the need for long-term antiretroviral treatment (ART). Those based on dendritic cells (DC) seems to be good at inducing HIV-specific T-cell responses and partial control of the viral load (VL) after antiretroviral therapy interruption (ATI) [1]. However, a lack of surrogate markers of the response means that the efficacy of HIV therapeutic vaccines is evaluated by the rebound in plasma VL after ATI.

There is an urgent need to develop markers that can accurately classify patients at a high or low probability of VL control after ATI. In recent years, omics techniques have improved, and transcriptome analysis can provide useful information for assessing vaccine efficacy [2]. In the HIV-1 vaccine field, it has been reported that a significant transcriptomic shift occurred in the peripheral blood mononuclear cells (PBMC) of individuals receiving a DC-based therapeutic vaccine [3]. Another study demonstrated the upregulation of certain genes related to direct cell recognition in the natural killer cells of healthy individuals who received a modified vaccinia Ankara (MVA) HIV-1 vaccine [4]. Changes in the gene expression of HIV-1-infected patients who received a DC therapeutic vaccine have been correlated with the post-vaccine peak VL [5].

There is increasing evidence about the influence of the microbiota on vaccine effectiveness [6,7], suggesting a direct link between a high diversity in the gut microbiota and a better vaccination response [8,9,10]. It is well-known that HIV infection reduces the richness and diversity of the intestinal microbe population and the depletion of some genera associated with anti-inflammatory conditions [11]. It is widely described that cohorts of people living with HIV (PLHIV), mainly represented by men who have sex with men, have a specific gut microbiota that is dominated by the *Prevotella* genus [12]. Additionally, elite controllers have presented differential relative abundances at the genus level that may be important in viral control [13]. In addition, there is evidence that the composition of the microbiota can predict the immune status in HIV infection [14].

The possibility that transcriptomics and metagenomics can be used as surrogate markers of the vaccine response has not been fully explored. In the present study, we aimed to identify the mRNA expression profiles and to assess the impact of a gut microbiome profile that can predict VL control after ATI in participants receiving a DC-based HIV-1 vaccine trial (NCT number: NCT02767193).

## 2. Materials and Methods

### 2.1. Patients and Study Design

A phase 2A randomized, double-blind, placebo-controlled clinical trial was performed in a single center. The study was approved by the institutional ethics review board and by the Spanish Regulatory Authorities (NCT number: NCT02767193).

Individuals with chronic HIV-1 infection were enrolled if they met the following criteria: age ≥ 18 years old, baseline CD4^+^ T-lymphocyte count > 450 cells/mm^3^, nadir CD4^+^ T-cell count > 350 cells/mm^3^, undetectable plasma VL (<50 copies/mL) for at least 6 months before recruitment, and on stable ART for at least 1 year before enrolment. These were randomized to receive three intranodal injections at 2-week intervals (i.e., 0, 2, and 4 weeks), alone or in combination with pegylated interferon α-2a (IFN). The randomization consisted of four arms in a 1:1:1:1 ratio, as follows:ARM I (*n* = 8): VACCINE. Autologous differentiated adult dendritic cells from monocytes of peripheral blood nonexpanded pulsed with autologous inactivated HIV virus.ARM II (*n* = 6): VACCINE + IFN. Autologous differentiated adult dendritic cells from monocytes of peripheral blood nonexpanded pulsed with autologous inactivated HIV virus with INF.ARM III (*n* = 7): PLACEBO. Autologous differentiated adult dendritic cells from monocytes of peripheral blood nonexpanded.ARM IV (*n* = 8): PLACEBO + IFN. Autologous differentiated adult dendritic cells from monocytes of peripheral blood nonexpanded with INF.

Twenty-nine patients were included and randomized into four different groups. Patients were followed up for 28 weeks after the first dose of vaccine (week 0). Each dose consisted of a concentration of ≥1 × 10^6^ autologous monocyte derived (MD)-DC pulsed with at least 10^9^ viral RNA copies (at the time of the pulse).

The dose used for the DC vaccine was obtained as demonstrated by F. Garcia et al., in which the DC vaccine provided an optimal immune boost [1]. On the other hand, peginterferon alfa-2b was administered once a week subcutaneously during weeks 4, 5, and 6 in those patients assigned to one of the branches that includes interferon administration. The dose of IFN depended on the patient’s weight and was dosed according to the manufacturer’s instructions. 

ART was interrupted (i.e., ATI) in all participants after the third injection of the vaccine or placebo. The consort statement describing the clinic trial was detailed by García F. et al. (manuscript under preparation).

PBMCs were collected from study participants 1 week before each vaccine dose. Monocytes were isolated from these samples and incubated with granulocyte-macrophage colony-stimulating factor and interleukin (IL)-4 at 37 °C for 5 days to induce cells to MD-DCs, which were used to produce the vaccines. The remaining monocyte-depleted PBMCs were frozen at −80 °C. Samples from week 3 were used in the present study (1 week before the last vaccine dose).

Patients with a VL decrease of >0.5 log_10_ copies/mL from the baseline (before any ART) to 12 weeks after ATI were considered “responders”, and all other patients were considered “non-responders”.

### 2.2. RNA Extraction, mRNA Library Preparation, and Sequencing

Total RNA was isolated from 5 × 10^6^ frozen monocyte-depleted PBMCs, using an miRNeasy® Mini Kit (Qiagen©, Hilden, Germany) according to the manufacturer’s instructions. Quality control was performed in a 4200 TapeStation System (Agilent, Santa Clara, USA). Samples with high quality were selected for downstream applications (RNA integrity number > 7) and conserved at −80 °C. The cDNA libraries from the total RNA samples were prepared with an Illumina TruSeq RNA sample prep kit (Illumina, San Diego, CA, USA) and clustered onto a TruSeq single-end flow cell using a TruSeq SR Cluster Kit v3-cBot-HS (Illumina, San Diego, CA, USA) after quantification with a PicoGreen dsDNA assay kit (Life Technologies, California, USA) and pooled in equimolar mixtures. Finally, the DNA sequence of each cluster on the flow cells was determined by employing 100 cycles of Sequencing-By-Synthesis (SBS) technology (TruSeq SBS Kit v3-HS kit) on an Illumina’s HiSeq2000 Sequencing System.

### 2.3. Stool Samples Collection, DNA Extraction, and Sequence

Fecal samples were collected from participants at weeks 0 and 5, according to the SOP.03.V2 protocol of the International Human Microbiome Standards. Samples were aliquoted and dry cryopreserved at −80 °C until DNA extraction. We followed the instructions in the handbook included with the QIAamp DNA stool mini kit. The Inhibitex Tablet (provided by the assay kit) was replaced by two additional steps with ammonium acetate 10M (Sigma-Aldrich, Milwaukee, WI, USA) and in a 1:1 mixture with isopropanol (Sigma-Aldrich, St. Louis, MO, USA). A denaturing step with glycoprotein denaturing buffer (New England Biolabs, Ipswich, MA, USA) and O/N incubation with 10-mg/mL Lysozime on 50-mM Tris-HCl at 37 °C were performed. Next, we used the following degenerate primers to amplify the conserved V3-V4 region of the 16 rRNA gene: Forward: 5′-ATT GAC GGG GRC CCG CAC-3; Reverse: 5′-CGA GCT GAC ARC CAT GCA-3′. Amplicon quality control was performed using a 5400 Fragment Analyzer System (Agilent, Santa Clara, CA, USA). Amplified DNA templates were cleaned of non-DNA molecules and Ilumina sequencing adapters. Sequencing was performed on an Illumina MiSeqTM platform (Illumina Inc., San Diego, CA, USA), according to the manufacturer’s specifications, to generate paired-end reads of approximately 300 base-pair lengths in each direction.

### 2.4. Bioinformatic and Statistical Analysis

The raw sequences generated by RNAseq were subjected to a quality control check using FastQC software (Babraham Inst, Cambridge, UK), and the filtered sequences were trimmed with Cutadapt [15] before alignment using the RSEM tool from R software (R Foundation for Statistical Computing, Vienna, Austria). An analysis of the differentially expressed genes was performed with the values normalized using the EBseq package included in the RSEM tool. To obtain a functional annotation of the obtained genes, we used the Gene Ontology and Kyoto Encyclopedia of Genes and Genomes (KEGG) pathways via the KOBAS platform [16].

The bacterial 16S rRNA data was generated using the QIIME2 software pipeline. Sequence quality control of generated raw sequences was assessed using the deblur package. A minimum read length of 190 base pairs was established, and chimeras and singletons were removed using the q2-uchime plugin. To assess the genus abundance, amplicon sequence variant counts were collapsed to the levels of the bacterial phylum and genus using the GreenGenes database. Downstream data analysis was performed with both QIIME2 and R software, using R packages vegan, *phylloseq*, *dplyr*, *magrittr*, and *ggpubr* for the statistical analysis and *ggplot2* for data plotting.

Comparison of parameters between groups and time points was performed in R. Differences between groups were assessed by multiple Mann–Whitney–Wilcoxon tests or Kruskal–Wallis tests with Bonferroni correction, as appropriate. We used two different approaches to study the effects of the intervention between the vaccine and placebo groups. First, we compared different time points (baseline vs. postvaccination) between groups. Second, we assessed the mathematical change (Δ) between the groups. The Δ was calculated for each marker as follows: Δ_Vaccine_ or Δ_Placebo_ = [value at the end of the study] − [value at baseline]

Microbiome samples were clustered by their genera composition, using nonmetric multidimensional scaling (NMDS) based on ecological distance matrices calculated by Bray–Curtis dissimilarities and implemented in R packages (Vegan, metaMDS, and ggplot2 packages). NMDS ellipses were drawn based on a confidence interval of 0.95.

Alpha diversity was determined using the following diversity indices (Shannon, inverse Simpson, and Fisher indices), together with richness parameters (observed). Beta diversity was determined by principal coordinates analysis (PCoA) with the obtained NMDS result and the gut microbiota composition (relative abundances). The differences in gut microbiota composition between groups were inspected by the linear discriminant analysis effect size (LEfSe). Sample dissimilarity between groups was evaluated using the Adonis (PERMANOVA) test.

Microbiome function was inferred using PICRUSt2 on GreenGenesDB-classified phylotypes. Counts were normalized to 16s rRNA gene copy numbers. To infer the gene content, the normalized phylotype abundances were multiplied by the respective set of gene abundances, represented by the KEGG identifiers estimated for each taxon.

## 3. Results

### 3.1. Patients

In total, we enrolled 29 patients living with HIV and randomized them to the intervention or placebo groups and further sub-grouped them by receipt of IFN therapy: the vaccine only group (ARM I; *n* = 8), the vaccine and IFN group (ARM II, *n* = 6), the placebo only group (ARM III, *n* = 7), and the placebo and IFN group (ARM IV; *n* = 8). The sample collection is summarized in Figure 1. No significant differences were detected in the plasma VL changes, HIV-specific T-cell responses, or reservoir features between the vaccine and placebo groups, which persisted when we compared those who received IFN (ARM II and ARM IV) with those who did not (ARM I and ARM III). Stopping the treatments (i.e., ATI) did not produce any alterations (not shown).

### 3.2. Vaccine Effect on the PBMC Transcriptome

To study the specificity of the DC vaccine transcriptome shift for chronic HIV infection, the data were compared with the placebo group. No differential gene expression was detected in patients before or after vaccinations when compared with the placebo group, regardless of the fold-change cut-off applied. Similarly, no significant features were detected in the blood transcriptome of immunized patients between the different time points (baseline and postvaccination) or in the placebo group (see Appendix A). These results suggest that the DC vaccine had no effect on the blood transcriptome.

### 3.3. Vaccine Effect on the Gut Microbiome

The gut microbiome changes induced by the DC vaccine were assessed by comparing the relative abundances of 16S rRNA data with the control group. One patient in the placebo group was excluded due to the poor quality of the stool sample received. No differences in richness and diversity (alpha diversity; Figure 2) were detected between the vaccine and placebo groups at the baseline and postvaccination. To characterize the inter-individual differences between groups (beta diversity), a PCoA was performed based on NMDS (Figure 3). The Adonis (PERMANOVA) test also yielded that the bacterial composition did not vary between the groups and different time points (*p* = 0.68, R^2^ = 0.04). The composition of the intestinal microbiota between the baseline and postvaccination data was examined by a LEfSe analysis, which revealed no differential bacterial features in the vaccine or control groups. These results suggested that the DC vaccine had no effect on the richness, diversity, or composition of the gut microbiota.

### 3.4. Post-Hoc Analysis: Response Effect

We conducted a further analysis considering the response effect (11 responders vs. 17 non-responders) irrespective of the treatment received. Grouping was done according to the required drop in VL indicating a response, as described in the Methods. We used both baseline and postvaccination data from each participant to ensure that we studied the effects of the gut microbiota composition and transcriptomics on the response behavior. For this purpose, we considered samples in each patient to be replicates, with each treated as a dependent variable in the statistical analysis. This avoided bias due to subject effect.

### 3.5. PBMC Transcriptome Analysis by Response Effect

An analysis of the differential gene expression was performed with the values normalized using the EBseq package. In total, 66 genes were expressed based on the set threshold level (fold-change (log_2_) ≥ 1 and *p*-value < 0.05). The comparison between responders and non-responders revealed that 31 of these were upregulated and 35 were downregulated in responders (Figure 4a). Functional annotation was obtained using KOBAS and the Gene Ontology (GO) and KEGG databases. The gene enrichment analysis revealed that 53 significant GO terms (Figure 4b) in the differentiation analysis were corrected for multiple comparisons (q-value < 0.05). However, upon inspection of the expression of biological processes, we found that the upregulated genes in responders were involved in the host defense (q = 0.001), type I interferon signaling (q = 0.001), apoptotic signaling (q = 0.01), positive regulation of T-helper 2 cell cytokine production (q = 0.03), viral process (q = 0.03), acute inflammatory response to antigenic stimulus (q = 0.03), and positive regulation of the immune response (q = 0.04). The differential expressed genes in the responder group are also involved in molecule transport processes, such as the regulation of protein export from the nucleus (q = 0.03) and mitochondrial membrane (q = 0.03) and the apoptotic signaling pathway (q = 0.01). Notably, three genes (i.e., IFI27, IFI6, and RSAD2) were upregulated in the above pathways in responders. Additional functional pathways with different distributions in the cohort are presented in Appendix A.

### 3.6. Gut Microbiota Analysis by Response Effect

A TGUT microbiome analysis was performed using the obtained relative abundances at the phylum and genus levels. Higher richness and diversity were detected in responders than in non-responders (Observed, *p* = 0.033; Fisher alpha, *p* = 0.021; Figure 5a). The LEfSe analysis revealed differences at the phylum level between groups: responders were enriched in *Bacteroidetes* (*p* < 0.005) and *Verrucomicrobia* (*p* = 0.017), while non-responders were enriched in *Tenericutes* (*p* = 0.049) and *Actinobacteria* (*p* < 0.005) (Figure 5b). We also found differences at the genus level between responders and non-responders in the LEfSe analysis. Responders showed enriched levels of *Bacteroides*, *Prevotella*, *Oribacterium*, *Methanosphaera*, *Bulleidia*, *Akkermansia*, and *Butyrivibrio*. By contrast, non-responders showed enriched levels of *Phascolarctobacterium*, *Mogibacterium*, and *Collinsella* (Figure 5c). Additionally, LASSO regression was performed to inspect the contribution of the response groups to the model (area under the receiver operating curve = 0.64).

### 3.7. Inferred Microbiome Functionality by Response Effect

The PICRUSt2 analysis to predict the metagenomic functional content of the gut microbiota revealed significant differences between the responder and non-responder groups at the KEGG level III. Specifically, the predicted pathway of amino acid metabolism for short-chain fatty acid (SCFA) production was increased significantly in the responders (Figure 6a), leading to L-leucine degradation, L-lysine fermentation to acetate and butanoate, L-methionine salvage cycle induction, and L-tyrosine degradation and ethylmalonyl-CoA pathway induction. In addition, an important increase in fatty acid biosynthesis was detected in responders that was absent in non-responders. Thus, the saturated and unsaturated fatty acid biosynthesis pathways were affected, including those of some long-chain fatty acids, such as palmitate (C16), oleate (C18), and stearate (C18) biosynthesis (Figure 6b).

## 4. Discussion

The transcriptomic regulation of the biological processes taking place during HIV infection has been investigated, but much remains unknown about this complex topic. Infection by HIV-1 leads to a host immune response that is both qualitatively and quantitatively different from that of other viral infections, with the virus targeting immune cells that are a major component of infection control. The consequence is an accumulation of dynamic proinflammatory and anti-inflammatory processes that are difficult to interpret; adding ART, ATI, and immunological interventions to the equation serves only to complicate matters further. The main goal of therapeutic vaccines for HIV-1 is to boost the immune system to control viral replication. De Goede et al. showed that a DC therapeutic vaccine caused a major transcriptomic shift in recipients by 5 weeks, whose shift was not further modified after ATI and the consequent viral rebound [3]. Moreover, they showed that most of this change was due to the activation of inflammatory and immune response pathways that remained essentially unchanged after ATI. Another recent publication showed that inflammatory pathways were downregulated during the first few weeks of a vaccination schedule with a similar DC therapeutic vaccine, followed by a significant upregulation by 12 weeks [5]. Unfortunately, we did not observe that the vaccination induced such changes. This could be due to the short follow-up time after administration of the last dose and makes it difficult to track the medium- and long-term effects of the intervention. Another possibility is that transcriptomic changes could be induced locally in the lymph node, with the transcriptomes of the PBMCs remaining unaffected [17].

We were also unable to identify vaccination-induced changes in the gut bacteriome. This may reflect the remoteness of the lymph node from the intestine and the short evaluation period after vaccination. The influence of the gut microbiota in vaccination responsiveness has been described [18,19,20], but the interplay with a DC vaccine is still unknown. It can be concluded that the intranodal administration of a DC vaccine did not affect the composition of the intestinal bacteria populations in the present study. Equally, we detected no impact on the gut microbiota 1 week after ATI, consistent with the existing literature [21].

This study evaluated the influence of the PBMC transcriptome and the gut microbiome on the efficacy of a DC therapeutic vaccine. Although the vaccine does not appear to affect the PBMC transcriptome or the gut microbiome, both omics data showed a relevant association with the VL response after ATI. Previously, our group described that the VL response to a DC therapeutic vaccine (manuscript under preparation) was related to inflammatory processes, such as tumor necrosis factor-alpha signaling, inflammatory response, and IL-6 and IL-2 signaling. The present work confirmed some of these findings, producing similar results in responders who presented the following: enriched levels of the IFI27 and IFI6 genes involved in the defense response to the virus, type I interferon (IFN) signaling, the positive regulation of T-helper 2 cell cytokine production, and an acute inflammatory response to antigenic stimulation. Our results also support previous data reporting that type I IFN activates a number of immune cells, including macrophages, DCs, and T cells [22]. This increased IFN signature has been reported to prevent early influenza virus replication in lung epithelia [23], and in cancer, the link between type I IFN and DCs is important for the antigen presentation to T cells [24]. Our results therefore highlight the importance of IFN pathways in HIV viral control, supporting the research stating that HIV infection triggers IFN responses in acute infections and contributes to HIV replication control [25]. The transcriptome analysis also revealed that the pathways associated with apoptotic signaling and mitochondrial metabolism were enriched in responders. Indeed, CD4+ T cells with inhibited apoptotic signals and mitochondrial dysfunction have been reported to be more susceptible to HIV-1 infection [26,27,28]. Further investigations are needed to determine the features of the transcriptome in individuals with lower VLs after ATI.

Several indices of the ecology of the microbiota in responders, such as richness and species count, were significantly higher than in non-responders. Furthermore, the responders had a unique bacterial signature of seven enriched genera. *Butyrivibrio*, SCFA-producing (butyrate) bacteria, are closely related to the homeostasis of the intestinal barrier integrity [29,30]. *Akkermansia* are associated with anti-inflammatory properties [31,32] and have been used as probiotics in obese mice, resulting in improved tissue inflammation [33]. They also help to maintain the gut barrier integrity, to modulate the immune response, to inhibit inflammation, and to promote colonization by butyrate-producing bacteria [34]. Finally, the responders were enriched in the two genera most often described in HIV (*Bacteroides* and *Prevotella*) [12], although their relative proportions were not altered.

The assessment of the predicted metabolic functions revealed that some functional pathways of amino acid metabolism were increased in responders. Included among these were L-lysine fermentation to acetate and butanoate, ethylmalonyl-CoA degradation, and L-tyrosine degradation. These three pathways can produce SCFA, directly or indirectly, as acetate, acetoacetate, and butyrate. It has been described that the end products of microbial fermentation, SCFAs, are key metabolites in the regulation of intestinal permeability and gut inflammation [35,36]. Consistent data exist on SCFA depletion in HIV-infected individuals receiving ART [37,38]. Interestingly, we described that greater long-chain fatty acid (LCFA) production by microbiota predicted metabolism functions in responders. Consistent with this, it has been shown that LCFAs could be degraded to acetoacetate via β-oxidation in the mitochondria [39,40]. We therefore hypothesize that responders have a functional gut microbiota that contributes to a delay in the viral rebound in HIV infection.

This study has several potential limitations. First, we evaluated the effect of the vaccination in the short term; longer follow-ups should be considered in similar studies. Second, the blood and feces samples were not collected simultaneously, making it difficult to associate the results obtained by the different omics approaches. Third, the low number of recruited individuals and the different interventions in each participant complicated the data interpretation further.

## 5. Conclusions

In conclusion, we could not discover any effect of the DC vaccine on the PBMC transcriptome and gut microbiota in the present study. However, an alternative analysis revealed that responders who had a longer time to viral rebound presented with an enrichment of the genes related to the type I IFN signaling pathway and were involved in the host defense response to the virus. Furthermore, the gut microbiota of the responders was richer, more diverse, and associated with the metabolic production of SCFA when compared with the non-responders.

## Figures and Tables

**Figure 1 vaccines-09-00694-f001:**
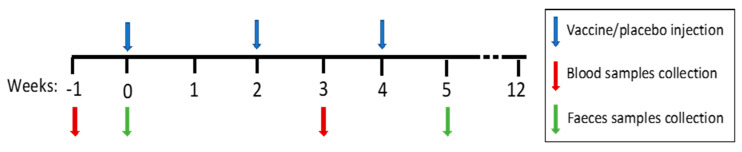
Study samples collection. Weeks −1 and 0 corresponded to the baseline of the study, and weeks 3 and 5 corresponded to the postvaccination points. Feces samples were collected at week 0 (baseline) and week 5 (postvaccination), indicated by the green arrows. Blood samples were collected at week −1 (baseline) and week 3 (postvaccination), indicated by the red arrows. The vaccine or placebo were injected at weeks 0, 2, and 4, indicated by the blue arrows.

**Figure 2 vaccines-09-00694-f002:**
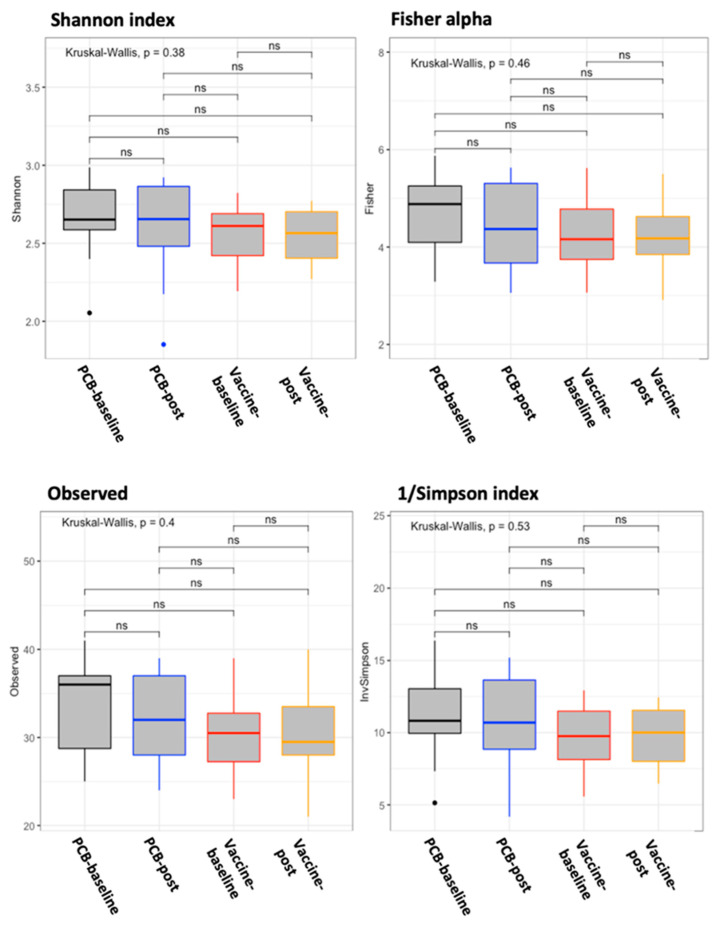
Alpha diversity by randomization group at different time points. Placebo (PCB) baseline (week 0) samples were represented in the black boxes, PCB post-vaccination in the blue boxes, vaccine baseline samples in the red boxes, and post-vaccination (post, week 5) samples in the orange boxes. Left-top: Shannon index, right-top: Fisher alpha index, left-bottom: number of observed OTUs, and right-bottom: inverse Simpson (1/Simpson) index. Box plots represent the median (black horizontal line), 25th, and 75th quartiles (edge of boxes) and upper and lower extremes (whiskers). For every parameter, the global Kruskal–Wallis *p*-value and the *p*-values for every comparison between the randomized groups at different time points are presented. *p*-Values code: *p* < 0.005: ***, *p* < 0.01: **, *p* ≤ 0.05: *, and *p* > 0.05: ns.

**Figure 3 vaccines-09-00694-f003:**
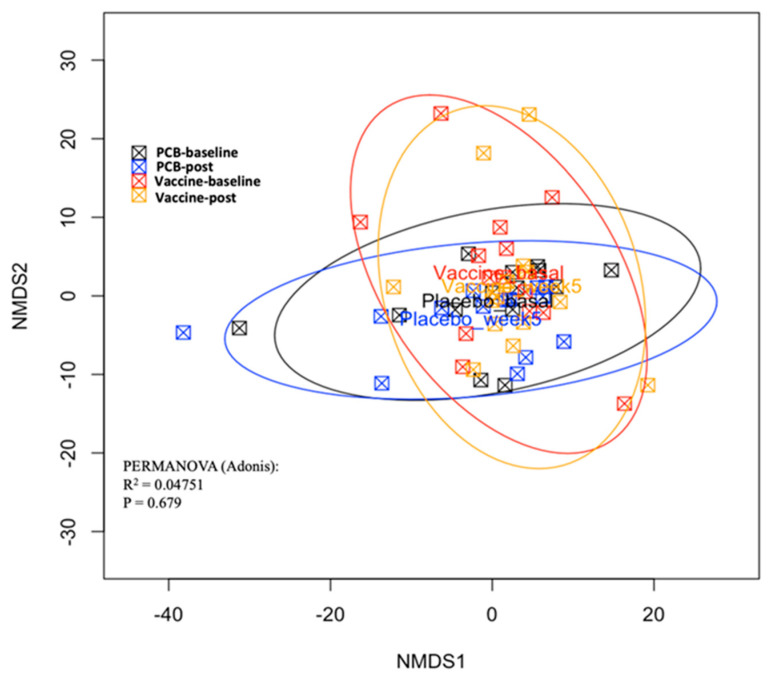
Principal coordinates analysis (PCoA) by randomization group. Placebo (PCB) baseline (week 0) samples were represented by the black squares, PCB post-vaccination (post, week 5) by the blue squares, vaccine baseline (week 0) samples by the red squares, and postvaccination (week 5) samples by the orange squares. The PCoA plot was obtained using the Bray–Curtis distances to visually analyse the beta diversity. These distances were based on dissimilarities in the microbiota profile of each subject. In the PCoA graph, each subject is shown as a square. The distances between the squares represent the differences in the microbiota composition between subjects. It is used to mathematically measure the involvement of groups in the beta diversity we performed in the Adonis test. *p*-Values code: *p* < 0.005: ***, *p* < 0.01: **, *p* ≤ 0.05: *, and *p* > 0.05: ns.It indicated that there were no differences in the beta diversity between the randomization groups (vaccine or placebo) at any time point analyzed. R package: vegan.

**Figure 4 vaccines-09-00694-f004:**
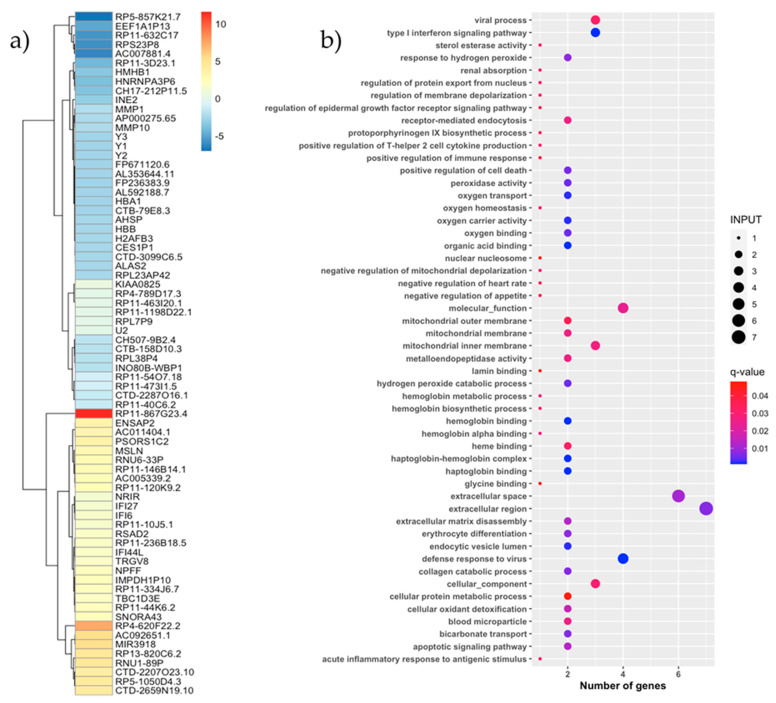
RNAseq transcriptomic analysis. (**a**) Heatmap of the fold change of the differential expressed genes between responder and non-responder individuals. The differential expressed gene symbol obtained from Ensembl genecode were showed in the vertical axis. The positive values (red scale colors) correspond to the upregulated genes in responder subjects, and the negative values (blue scale colors) correspond to the downregulated genes in responder subjects. The differential expressed genes were obtained using the ANOVA test, and a *p*-value < 0.05 was considered significant. (**b**) Enrichment analysis of the differential expressed genes in the responder group. The functional annotation was obtained using KOBAS software and the Gene Ontology (GO) database. The differential expressed pathways with q-value < 0.05 were considered significant. Patients were considered as responders with a delta VL set point >0.5 log10 copies/mL.

**Figure 5 vaccines-09-00694-f005:**
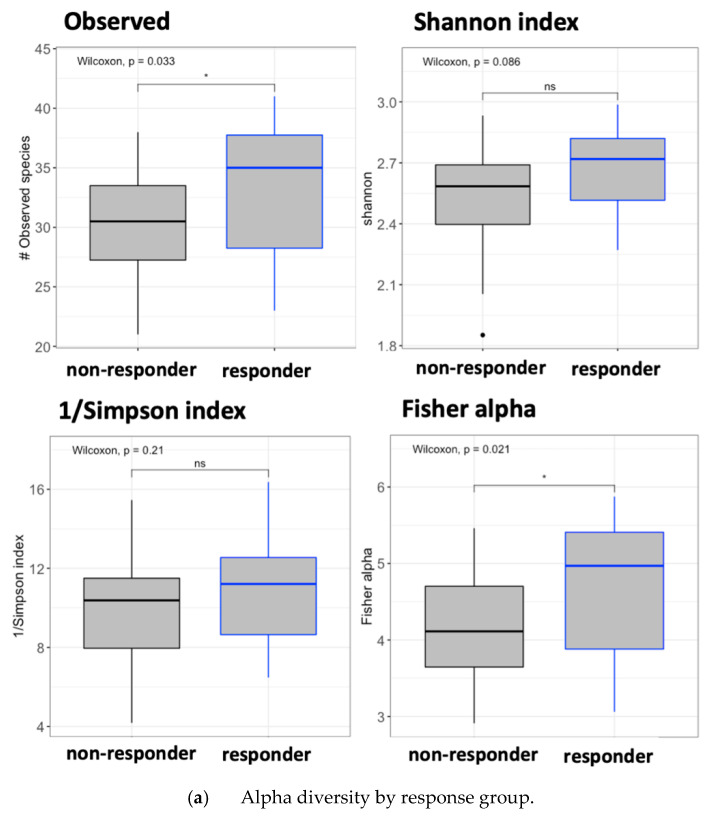
Gut microbiota composition analysis by response effect. (**a**) Alpha diversity by response effect group. Black boxes represent the non-responder group, and blue boxes represent the responder group. Left-top: number of observed OTUs, right-top: Shannon index, left-bottom: Fisher alpha index, and right-bottom: Simpson index. Box plots represent the median (black horizontal line), 25th, and 75th quartiles (edge of boxes) and upper and lower extremes (whiskers). Outliers are represented by a single data point. For every parameter, the global Kruskal–Wallis *p*-value and the *p*-values for every comparison between the randomized groups at different time points are presented. *p*-Values code: *p* < 0.005: ***, *p* < 0.01: **, *p* ≤ 0.05: *, and *p* > 0.05: ns. (**b**) LefSe analysis to detect the enriched bacteria at the phylum (p_) level between responder and non-responder subjects. (**c**) LefSe analysis to detect the enriched bacteria at the genus (g_) level between responder and non-responder subjects. Patients were considered as responders with a delta VL set point >0.5 log10 copies/mL.

**Figure 6 vaccines-09-00694-f006:**
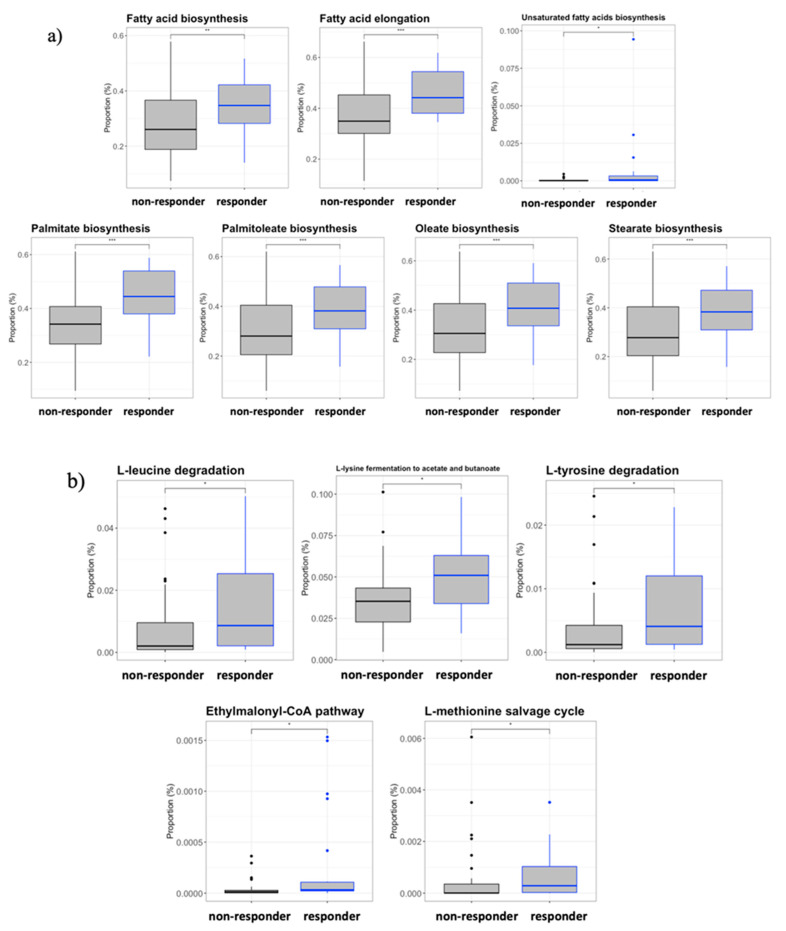
Inferred functional content of the gut microbiota. The metagenomic functional content of the gut microbiota was predicted by the inferred PICRUSt2 plugin from QIIME2 software. (**a**) Amino acid degradation and metabolism pathways. (**b**) Fatty acid biosynthesis and metabolism pathways. Patients were considered as responders with a delta VL set point >0.5 log10 copies/mL. For every parameter, the Wilcoxon test *p*-value for every comparison between the randomized groups at different time points are presented. *p*-Values code: *p* < 0.005: ***, *p* < 0.01: **, *p* ≤ 0.05: *, and *p* > 0.05: ns.

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
