# Peer review of "Impact of Transcriptome and Gut Microbiome on the Response of HIV-1 Infected Individuals to a Dendritic Cell-Based HIV Therapeutic Vaccine"

_vaccines, 2021, doi:10.3390/vaccines9070694_

Round 1

Reviewer 1 Report

This research article reports on clinical data obtained from a small Phase 2A clinical trial to study the HIV load reduction on effect on transcriptome and gut microbiome after administration with DC based HIV vaccine. Analysis of the results indicate DC vaccine had no effect on the richness, diversity and composition of the gut microbiota. But analysis found enriched genes of IFN signaling  pathway and metabolic production of SCFA.

Major/Minor issues:

  1. Study has only 29 patients, which is a very small number to get any relevant data.
  2. Expand SCFA in abstract.
  3. Occasional typos can be avoided by careful proof reading.

As authors themselves acknowledged this study has number of limitations: 1) the effect of vaccination was assessed in the short term, 2) the blood and feces samples were not collected simultaneously making it difficult to associate the results obtained, and 3) the low number of recruited individuals and the different interventions tested on the enrolled subjects make it difficult the data interpretation.

In summary this study was not able to discover any effect of the DC vaccine on the PBMC transcriptome and the gut microbiota. But an alternative analysis characterized that responder individuals with a longer time to viral rebound, presented enriched genes of type I interferon (IFN) signaling pathway involved in defense response to virus. Furthermore, the gut microbiota of these responder individuals was richer and more diverse than non-responder subjects. This microbiota from responder individuals were related with the metabolic productions of short chain fatty acid.

Author Response

Reviewer #1:

This research article reports on clinical data obtained from a small Phase 2A clinical trial to study the HIV load reduction on effect on transcriptome and gut microbiome after administration with DC based HIV vaccine. Analysis of the results indicate DC vaccine had no effect on the richness, diversity and composition of the gut microbiota. But analysis found enriched genes of IFN signalling pathway and metabolic production of SCFA.

Thanks to the reviewer #1 for his kind words and suggestions. We address all the concerns of the referees here. Our revisions reflected all reviewers’ suggestions. The added text to the main manuscript was highlighted in blue colour. Detailed responses to reviewers are given below.

Major/Minor issues:

  1. Study has only 29 patients, which is a very small number to get any relevant data.

We understand the reviewer's comment on the study's limitation of the limited number of participants. This is due to the complex process of recruiting people living with HIV on stable antiretroviral therapy and sorting participants by gender, age, sexual behaviour and other relevant characteristics.

  1. Expand SCFA in abstract.

Done. Included at the end of the abstract (line 55-58):

“SCFA are key metabolites in the regulation of intestinal homeostasis. There exists consistent information on SCFA depletion in HIV-infected individuals on ART.”

  1. Occasional typos can be avoided by careful proof reading.

A native English speaker has checked the whole manuscript grammar to solve the language problem observed by the reviewer.

As authors themselves acknowledged this study has number of limitations: 1) the effect of vaccination was assessed in the short term, 2) the blood and faeces samples were not collected simultaneously making it difficult to associate the results obtained, and 3) the low number of recruited individuals and the different interventions tested on the enrolled subjects make it difficult the data interpretation.

In summary this study was not able to discover any effect of the DC vaccine on the PBMC transcriptome and the gut microbiota. But an alternative analysis characterized that responder individuals with a longer time to viral rebound, presented enriched genes of type I interferon (IFN) signalling pathway involved in defense response to virus. Furthermore, the gut microbiota of these responder individuals was richer and more diverse than non-responder subjects. This microbiota from responder individuals were related with the metabolic productions of short chain fatty acid.

Reviewer 2 Report

The paper by Pastor-Ibanez et al needs an extensive editing of English language and style, as well as references that completely missing the titles of the articles. The paper is difficult to read and the results doesn’t convince and support enough the conclusion.

Figure 3 not clear, Figure 5 incorrect.

I personally suggest to submit the work to a journal more specialized in proteomic and transcriptomic analyzes.

Author Response

Reviewer #2:

The paper by Pastor-Ibanez et al needs an extensive editing of English language and style, as well as references that completely missing the titles of the articles. The paper is difficult to read and the results doesn’t convince and support enough the conclusion.

Thanks to the reviewer #2 for his comments. We address all the concerns of the referees here. Detailed responses to reviewers are given below.

We appreciate the criticism about the language and style. In order to solve the difficulties in the text reading detected by the reviewer #2, we contacted a native English speaker to carefully corrected the language errors. Additionally, we have checked and corrected the errors in the references noted by the reviewer.

  • References that completely missing the titles of the articles

Thank you to the reviewer to detected the error in the reference #23. We have corrected it in the main manuscript (line 503-504).

We are grateful to receive the reviewer comments about the figures of this work.

  • Figure 3 not clear

In order to clarify the Figure 3, we detailed the presented content below. It has been included in the footnotes to Figure 3:

The Figure 3 represent de Principal Coordinates Analysis (PCoA) obtained using the Bray-Curtis distances to visually analyse beta-diversity. These distances were based on dissimilarities in the microbiota profile of each subject. In the PCoA graph, each subject is shown as a square. The distances between the squares represent the differences in microbiota composition between subjects. It is used to mathematically measure the involvement of groups in the beta-diversity we performed in the Adonis test. Figure 3 indicated that there were no differences in beta-diversity between the randomization groups (Vaccine or Placebo) at any time-point analysed.

  • Figure 5 incorrect.

As the reviewer noted, there exist a typo erroring the numeration of the subplots included in the Figure 5 (a, b and c). We have corrected it in the main manuscript.

  • I personally suggest to submit the work to a journal more specialized in proteomic and transcriptomic analyses.

We are grateful to receive the reviewer suggestion. In this respect, we are convinced that Vaccines is the ideal journal in which to present this work, where the effect of a vaccine against the HIV infection on the gut microbiota and blood transcriptomic was described. Unhopefully, this study could not find any vaccine-induced changes on both microbiota and transcriptomic profiles. However, this study revealed a special composition of gut bacteria and blood transcriptomics in subjects with a longer time to viral rebound after ART interruption.

Reviewer 3 Report

Pastor-Ibáñez and colleagues report on the omics analysis of a individuals in a clinical trial of a DC-based HIV vaccine in combination with IFNalpha2a. The need to better treat and eventually to proactively vaccinate against HIV infection remains an urgent matter. Thus, the manuscript addresses an interesting topic. However, the rationale for the design of the clinical trial and the present study remains largely unclear. It is known that treatment with IFNalpha has only a short-term antiviral effect and that prolonged exposure to IFNalpha eventually leads to a burn-out effect and compromises a systemic immune response. What was the rationale to combine a DC-based vaccine with IFNalpha2a? How was the dosage for DC-based vaccine and IFNalpha decided? Related to that question, in Materials and Methods: a sentence states ‘… heat-inactivated autologous virus (109 RNA copies/dose …’. Was the dose 10E9 or 109 copies? Back to the study in its entirety, what was the rationale for the different timing of blood and microbiome collection and why weren’t any blood samples analyzed after 5 weeks or at 12 weeks?

The following statements in the Results section are confusing because and even though the authors explain that the analysis grouping the study subjects by responder versus non-responder reveals differences: ‘… results suggest that the DC vaccine had no effect on the blood transcriptome.’, and ‘…DC vaccine had no effect on the richness, diversity and composition of the gut microbiota.’. However, it is surprising that no effect of IFNalpha2a is detected on blood transcriptome without grouping by responders and non-responders based on differences on VL after ATI. What is the explanation for that? Was the IFNalpha2a dosage too low? One would expect to see a difference at least between ARM IV  and ARMIII.

The findings for the microbiome and SCFA in association with the responder phenotype are interesting and warrant further investigation.

Author Response

Reviewer #3:

Pastor-Ibáñez and colleagues report on the omics analysis of a individuals in a clinical trial of a DC-based HIV vaccine in combination with IFNalpha2a. The need to better treat and eventually to proactively vaccinate against HIV infection remains an urgent matter. Thus, the manuscript addresses an interesting topic. However, the rationale for the design of the clinical trial and the present study remains largely unclear. It is known that treatment with IFNalpha has only a short-term antiviral effect and that prolonged exposure to IFNalpha eventually leads to a burn-out effect and compromises a systemic immune response. What was the rationale to combine a DC-based vaccine with IFNalpha2a? How was the dosage for DC-based vaccine and IFNalpha decided? Related to that question, in Materials and Methods: a sentence states ‘… heat-inactivated autologous virus (109 RNA copies/dose …’. Was the dose 10E9 or 109 copies? Back to the study in its entirety, what was the rationale for the different timing of blood and microbiome collection and why weren’t any blood samples analyzed after 5 weeks or at 12 weeks?

The following statements in the Results section are confusing because and even though the authors explain that the analysis grouping the study subjects by responder versus non-responder reveals differences: ‘… results suggest that the DC vaccine had no effect on the blood transcriptome.’, and ‘…DC vaccine had no effect on the richness, diversity and composition of the gut microbiota.’. However, it is surprising that no effect of IFNalpha2a is detected on blood transcriptome without grouping by responders and non-responders based on differences on VL after ATI. What is the explanation for that? Was the IFNalpha2a dosage too low? One would expect to see a difference at least between ARM IV and ARMIII.

The findings for the microbiome and SCFA in association with the responder phenotype are interesting and warrant further investigation.

We grateful to reviewer #3 for considering this study as an interesting topic. In order to solve the questions raised by the reviewer, we detailed the rationale of the design of this clinical trial. The added text was highlighted in blue colour in the main manuscript.

  • The rationale for the design of the clinical trial and the present study remains largely unclear

Regarding the schedule of the study design, we detailed the complete procedure used in this clinical trial.

This was single-centre, national clinical trial, phase I, randomized (1: 1: 1: 1), prospective, placebo-controlled, partially masked, parallel group. Patients will be assigned to one of the following four arms:

  1. ARM I: VACCINE. Autologous differentiated adult dendritic cells from monocytes of peripheral blood non expanded pulsed with autologous inactivated HIV virus.
  2. ARM II: VACCINE + IFN. Autologous differentiated adult dendritic cells from monocytes of peripheral blood non expanded pulsed with autologous inactivated HIV virus with INF.
  • ARM III: PLACEBO. Autologous differentiated adult dendritic cells from monocytes of peripheral blood non expanded.
  1. ARM IV: PLACEBO + IFN. Autologous differentiated adult dendritic cells from monocytes of peripheral blood non expanded with INF.

We included it in the Methods section (line 120-129)

The complete information about the clinical trial could be founded in https://clinicaltrials.gov/, ClinicalTrials.gov Identifier: NCT02767193.

We understand the reviewer’s comment about the study design of the clinical trial. It is remarkable that our group is now working to describe the complete clinical outcomes of this clinical trial in a different manuscript, so the graphs about the study design have been reserved for the cited work.

  • It is known that treatment with IFNalpha has only a short-term antiviral effect and that prolonged exposure to IFNalpha eventually leads to a burn-out effect and compromises a systemic immune response. What was the rationale to combine a DC-based vaccine with IFNalpha2a?

We appreciate the reviewers’ questions about the combined administration of a DC vaccine with IFN alpha. A recently published study looked at the effect of IFN in HIV-infected patients on cART [1]. In these patients, a decrease in the viral reservoir was observed, even after cART interruption, as measured by the proviral DNA technique. In addition, it has been observed that interferon alpha/beta can increase CD8+ T-cell responses [2, 3]. Its combination with dendritic cell-based therapeutic vaccines has been tested in cancer, where its use has been found to be safe and to enhance antigen-specific immune responses [4, 5]. We hypothesise that the combination of a DC vaccine with pegylated interferon may help to decrease the reservoir, enhance antigen presentation and divert the immune response to a TH1 response that favours a stimulation of HIV-specific CD8 responses through more effective cross-presentation [2].

References:

  1. Azzoni L, Foulkes AS, Papasavvas E, Mexas AM, Lynn KM, Mounzer K, et al. Pegylated Interferon alfa-2a monotherapy results in suppression of HIV type 1 replication and decreased cell-associated HIV DNA integration. J Infect Dis 2013; 207(2):213-222
  2. Le BA, Etchart N, Rossmann C, Ashton M, Hou S, Gewert D, et al. Cross-priming of CD8+ T cells stimulated by virus-induced type I interferon. Nat Immunol 2003; 4(10):1009-1015.
  3. Longhi MP, Trumpfheller C, Idoyaga J, Caskey M, Matos I, Kluger C, et al. Dendritic cells require a systemic type I interferon response to mature and induce CD4+ Th1 immunity with poly IC as adjuvant. J Exp Med 2009; 206(7):1589-1602.
  4. Schwaab T, Schwarzer A, Wolf B, Crocenzi TS, Seigne JD, Crosby NA, et al. Clinical and immunologic effects of intranodal autologous tumor lysate-dendritic cell vaccine with Aldesleukin (Interleukin 2) and IFN-{alpha}2a therapy in metastatic renal cell carcinoma patients. Clin Cancer Res 2009; 15(15):4986-4992.
  5. Alfaro C, Perez-Gracia JL, Suarez N, Rodriguez J, Fernandez de SM, Sangro B, et al. Pilot clinical trial of type 1 dendritic cells loaded with autologous tumor lysates combined with GM-CSF, pegylated IFN, and cyclophosphamide for metastatic cancer patients. J Immunol 2011; 187(11):6130-6142.

  • How was the dosage for DC-based vaccine and IFNalpha decided?

The dose used for the DC vaccine was obtained as F. Garcia et al [6] in which the DC vaccine provided an optimal immune boost. In the other side, Peginterferon alfa-2b was administered once a week subcutaneously during weeks 4, 5 and 6. In those patients assigned to one of the branches that includes interferon administration. The dose depended on the patient's weight and was dosed according to the instructions in the data sheet.

We included it in the Methods section (line 135-140)

The complete information about the clinical trial could be founded in https://clinicaltrials.gov/, ClinicalTrials.gov Identifier: NCT02767193.

  1. García F, Climent N, Assoumou L, Gil C, González N, Alcamí J, et al. A therapeutic dendritic cell-based vaccine for HIV-1 infection. J Infect Dis. 2011;203(4):473–8.

  • Related to that question, in Materials and Methods: a sentence states ‘… heat-inactivated autologous virus (109 RNA copies/dose …’. Was the dose 10E9 or 109 copies?

As the reviewer states, we stablished a minimum concentration of heat-inactivated autologous HIV virus of 109 copies to load into the autologous DC. So, the final dose injected consist in a concentration of ≥1x106 autologous DC pulsed with at least 109 viral RNA copies.

In order improve understanding of the study design, we have included this clarification in the main manuscript in lines 132-134:

“Each dose consisted of a a concentration of ≥1x106 autologous monocyte derived (MD)-DC pulsed with at least 109viral RNA copies (at the time of the pulse).”

  • Back to the study in its entirety, what was the rationale for the different timing of blood and microbiome collection and why weren’t any blood samples analysed after 5 weeks or at 12 weeks?

We really appreciate the reviewer's considerations. The reason to collect the blood and stool samples at different time-points lies on the lack of availability to obtain the blood samples. There was no possibility to obtain additional blood samples. In order to state this issue, we included this in the limitations section (line 417-422). Regarding the blood samples collection, we agree with the reviewer question about the short follow-up of the transcriptomics shifts induced by the vaccination, but we followed the same strategy in the transcriptomics analysis as in similar studies performed in our centre [6]. This, coupled with the lack of experience in the approach to microbiota studies by the medical team, limited the conclusions of the clinical trial.

  • However, it is surprising that no effect of IFNalpha2a is detected on blood transcriptome without grouping by responders and non-responders based on differences on VL after ATI. What is the explanation for that? Was the IFNalpha2a dosage too low?

We welcome the reviewer’ comment on the effect of the IFN alpha on the blood transcriptome. Unfortunately, it we were not able to analyse this effect because the first dose of IFN was at week 4 and the last blood samples collected for transcriptomic analysis were at week 3. In the case of the effect of the IFN ono the gut microbiota, we did not observe any shift on it, this could be due to the short time elapsed between the first dose of IFN (week 4) and the last stool sample collection (week 5). It is true that longer follow-up could reflected the IFN impact on the immunity and the microbiota populations and we apologize for the lack of this information.

  • One would expect to see a difference at least between ARM IV and ARMIII.

We appreciate the reviewer's interest in the submitted work. We hypothesised that ARM III serves as a double negative control for the study, meaning that ARM III reflects the stimulation of the immune system by DCs not pulsed with virus and without the effect of IFN. On the other hand, ARM IV was considered as the control for the adjuvant effect of IFN on the immune system drive.

Moreover, both ARM III and IV were useful to test the cross-reactivity between the different compounds used in the clinical trial.

  • The findings for the microbiome and SCFA in association with the responder phenotype are interesting and warrant further investigation.

We agree with the reviewer’ consideration about the SCFA implications in responder subjects. Unfortunately, we were unable to measure and analyse these metabolites due to economical reasons. We will consider to analyse metabolomics shifts in the upcoming microbiome studies.

Round 2

Reviewer 2 Report

  Thanks to the authors for accepting and discussing the suggestions provided.
At the moment, the manuscript is sufficiently improved and is considered suitable for publication.

Reviewer 3 Report

The authors have well addressed the comments of the previous review.